# The Culturable Mycobiota of Sediments and Associated Microplastics: From a Harbor to a Marine Protected Area, a Comparative Study

**DOI:** 10.3390/jof8090927

**Published:** 2022-08-31

**Authors:** Matteo Florio Furno, Anna Poli, Davide Ferrero, Federica Tardelli, Chiara Manzini, Matteo Oliva, Carlo Pretti, Tommaso Campani, Silvia Casini, Maria Cristina Fossi, Giovanna Cristina Varese, Valeria Prigione

**Affiliations:** 1Department of Life Sciences and Systems Biology, University of Torino, Mycotheca Universitatis Taurinensis (MUT), Viale Mattioli 25, 10125 Torino, Italy; 2Interuniversity Consortium of Marine Biology and Applied Ecology “G. Bacci” (CIBM), Viale N. Sauro 4, 57128 Livorno, Italy; 3Department of Physical, Earth and Environmental Sciences, University of Siena, Via Mattioli 4, 53100 Siena, Italy

**Keywords:** marine fungi, Mediterranean Sea, microplastics, sediment, plastisphere, fungal community

## Abstract

Fungi are an essential component of marine ecosystems, although little is known about their global distribution and underwater diversity, especially in sediments. Microplastics (MPs) are widespread contaminants worldwide and threaten the organisms present in the oceans. In this study, we investigated the fungal abundance and diversity in sediments, as well as the MPs, of three sites with different anthropogenic impacts in the Mediterranean Sea: the harbor of Livorno, the marine protected area “Secche della Meloria”; and an intermediate point, respectively. A total of 1526 isolates were cultured and identified using a polyphasic approach. For many of the fungal species this is the first record in a marine environment. A comparison with the mycobiota associated with the sediments and MPs underlined a “substrate specificity”, highlighting the complexity of MP-associated fungal assemblages, potentially leading to altered microbial activities and hence changes in ecosystem functions. A further driving force that acts on the fungal communities associated with sediments and MPs is sampling sites with different anthropogenic impacts.

## 1. Introduction

Marine sediments cover ~70% of the Earth’s surface. These environments are largely unexplored and harbor a broad diversity of microorganisms [1], which have become objects of increasing interest. Several studies have shown the crucial role of sediment fungal communities in fundamental ecosystem processes (e.g., biogeochemical and food-web dynamics or degradation of organic matter) [2,3]. However, in comparison to other environmental matrices such as soil and seawater [4,5], the fungal diversity associated with marine sediments remains poorly characterized [4,5]. Moreover, most of the studies that analyzed the marine sediment fungal communities focused on peculiar environments such as deep-sea and benthic habitats [6,7,8], oxygen-deficient sediments, or hydrothermal vents [9,10], while the mycobiota associated with more accessible shallow-water sediments, such as those of coastal areas, are poorly investigated [2,9,10].

The majority of the studies focusing on sediment fungi are based on culture-independent approaches. Despite their advantages in detecting unculturable organisms and in hinting at functional traits, culture-independent methods may obscure the value of fungal isolation in axenic conditions. Indeed, culturomics gives the opportunity to assume an ecological role and to exploit the strains isolated for future biotechnological applications, due to the adaptation to these extreme habitats. For instance, culture-independent approaches are almost useless when dealing with problems related to marine sediments, such as the bioremediation of heavily-polluted sediments [11,12].

Today, there are various problems related to marine sediments [13,14,15]. In particular, one of the main issues is the impact of the anthropic activities that threaten these ecosystems, especially in populous coastal areas. Lately, researchers’ attention has focused on microplastics (MPs) [16,17,18]: once released in the sea, MPs, transported by currents, threaten the organisms present in the oceans [19] and eventually accumulate in the sediments and water columns [18]. The estimated total amount of floating plastic (and MPs) debris only accounts for less than 1% of the total estimated plastic input [20], indicating the presence of extensive MPs sinks in marine sediments, especially coastal ones [21]. Indeed, MP concentration is often closely related to human population density (e.g., the presence of wastewater treatment plants), indicating that nearshore zones are an important source [21]. In addition, MPs represent a new ecological niche for microorganisms in the marine habitat (including invasive species and/or pathogens): the “plastisphere” [22]. Most of the studies that have investigated the ecology and diversity of “MPs microorganisms” mainly focused on bacteria and algae. Fungi, instead, are generally neglected [23,24]. 

In this study, we intended to investigate the culturable fungal diversity inhabiting sediments and associated MPs sampled in three areas of the Tyrrhenian Sea in Tuscany (IT): (i) the harbor of Livorno (LH), (ii) the Marine Protected Area “Secche della Meloria” (MPA), and (iii) an intermediate point (IP) (Figure 1). The three areas have different anthropic impacts: various petrochemical and industrial activities have influenced LH, mainly through metal and hydrocarbon contamination [25], whereas the MPA is an area of more than 900 ha, located three miles from LH and with an exceptional historical, archaeological, and natural interest [26]; IP, represents an intermediate point along the transect.

## 2. Materials and Methods

### 2.1. Sample Collection and Processing

Marine sediments and MPs were collected in November 2019 by scuba divers of the “Interuniversity Consortium of Marine Biology G. Bacci, CIBM” of Livorno, using a manual core drill, in three areas of the Tyrrhenian Sea in Tuscany (IT): the Yacht Club in LH (43°33′00.1″ N 10°17′51.8″ E), the MPA “Secche della Meloria” (43°32′ 46.08″ N 10°13′ 8.62″ E), and at the IP (43°54′64.9″ N 10°26′21.1″ E) (Figure 1). Sediments were collected from three sub-sites (A, B, C) in each site. For MPs a composite sample from the three sub-sites of each site was performed.

The collected sediments were left to decant, to separate the watery fraction (WS) from the solid one (SS). Therefore, each sample consisted in two 1 L clean plastic jars filled with SS and one 0.5 L clean plastic jar filled with WS. MPs were extracted from the sediments by density separation, using insufflation of air [27,28]. For each site, MPs (about 15–20 fragments) were washed with sterile seawater and placed in a sterile falcon containing 30 mL of sterile seawater. The two fractions of sediments and MPs were stored at 4 °C.

### 2.2. Sediment and MPs Characterization 

For each site, a representative sample of sediment and MPs was collected for characterization. 

Sediment samples were analyzed for: (i) grain size classification of sediments, (ii) trace elements quantification, and (iii) organopollutants quantification.

(i)Grain size classification was performed on sediment samples, following Italian Superior Institute for Environmental Protection (Istituto Superiore per la Protezione Ambientale—ISPRA) guidelines [29].(ii)For trace element quantification, Hg standard stock solutions were prepared in HNO_3_ 2% (concentration range: 5–200 μg/L). Before metal analyses, samples were dried at 40 °C for 48 h and sieved through a 2-mm mesh size sieve. Dried samples were directly analyzed, to evaluate the level of Hg according to EPA 7473 [30]. As regards the other trace element quantifications, sediments were digested following EPA 3051A [31]. Al, Cr, Cu, Fe, Ni, V, and Zn analyses were performed with an ICP Varian 720-ES (Agilent), according to EPA 6010D [32]; while As, Cd, and Pb were quantified with an AAS Varian SpectrAA240Z, following EPA 7010 [33].(iii)Polycyclic aromatic hydrocarbons (PAHs as total content and 16 EPA congeners), and polychlorinated biphenyls (PCBs as total content) were extracted and processed according to the methods EPA3545A [34] (extraction) and EPA 8270E [35] (analysis). Low molecular weight hydrocarbons (C < 10) were determined following the methods EPA5021A [36] (extraction with headspace technique) and EPA 8015C [37] (GC analysis). High molecular weight hydrocarbons (C > 10) were extracted and measured according to the method ISO 16703 [38].

The extraction and characterization of MPs from sediments samples was conducted following the JPI-oceans method [28]. Briefly, MPs were extracted using high density separation, using a saturated sodium chloride solution (NaCl—density: 1.2 g cm^−3^). Subsequently, the supernatant was filtered using a vacuum pump filtration kit, using a 63-μm metal mesh filter. The obtained filters from each sample were observed under a stereomicroscope with a magnification of 6.5X, in order to characterize MPs according to shape, type, and color.

### 2.3. Fungal Isolation

Depending on the matrix, different methodologies were applied: (i) sonication and direct plating for MPs; (ii) soil dilution plate for the solid fraction of the sediments; and (iii) filtration for the watery fraction of the sediments. To maximize the number of cultivable fungi, isolation was carried out using three culture media: malt extract agar (MEA: 20 g L^−1^ malt extract, 20 g L^−1^ glucose, 2 g L^−1^ peptone, 18 g L^−1^ agar); corn meal agar (CMA: 17 g L^−1^ CMA extract, 18 g L^−1^ agar); synthetic nutrient-poor agar (SNA: 1 g L^−1^ KH2PO4, 1 g L^−1^ KNO3, 0.5 g L^−1^ MgSO4. 7H2O, 0.5 g L^−1^ KCl, 0.2 g L^−1^ glucose, 0.2 g L^−1^ saccharose, 18 g L^−1^ agar). Each medium was supplemented with antibiotics (Gentamicin 80 mg L^−1^ and Tazobactam 100 mg L^−1^) and 2% *w*/*v* of sea salts. All the chemicals were purchased from Sigma-Aldrich (St. Louis, MO, USA).

MPs were sonicated for 3 min and vortexed for 1 min, to detach hyphae and conidia (repeated 3 times). Subsequently, inoculation was carried out by spreading 1 mL of supernatant in a Petri dish (15 cm ⌀). Ten replicates per medium and per sample were performed. Moreover, ten MPs for each site were recovered with sterile tweezers and placed directly onto a Petri dish (15 cm ⌀), to allow the isolation of fungi still stuck to the surface of the MPs. 

For SS, a modified version of the method of Nasrawi et al. [11] was applied. First, due to the heterogeneity of the sediments, the samples were sifted through a 2-mm sieve to make them comparable. An amount of wet sediment, corresponding to 10 g of dry weight of each sample, was suspended in 90 mL of sterile saline solution NaCl 0.9% *w*/*v*. Following a preliminary test [39] the final dilution (1:10 for MPA and IP, 1:100 for LH) was plated (1 mL per plate) onto Petri dishes (15 cm ⌀). For WS, the method described by Bovio et al. [40] was applied. Briefly, 30 aliquots of 10 mL of WS sample were filtered through sterile nitrocellulose membranes (50 mm ⌀, 0.45 μm pore, VWR), which were then transferred onto 9-cm ⌀ Petri dishes containing the abovementioned culture media. Ten replicates per medium and per sample were performed. 

A total of 660 plates were incubated in the dark at 15 °C (average temperature of seabed), and periodically inspected up to 30 days. Colony forming units per gram of dry weight (CFU g^−1^ dw) for sediment and per 100 mL (CFU 100 mL^−1^) of filtered water were recorded. The relative abundance (RA%) was calculated as the number of CFU for each species out of the total load. Isolated fungi were maintained in pure culture.

### 2.4. Fungal Identification

Fungi were identified using a polyphasic approach, combining macro- and microscopic features with molecular analyses. First, fungi were identified to genus level on the basis of their morphology, allowing a choice of the most appropriate primers for the amplification of specific genetic markers. Next, genomic DNA was extracted using a NucleoSpin kit (Macherey Nagel GmbH, Duren, DE, USA), according to the manufacturer’s instructions. As described by Garzoli et al. [41], specific markers were amplified in a Biometra TGradient Thermocycler (Biometra, Gottingen, Germany). The PCR mixture consisted of 5 μL 10x PCR Buffer (15 mM MgCl2, 500 mM KCl, 100 mM Tris-HCl, pH 8.3) 0.4 mM MgCl2, 0.2 mM each dNTP, 1 μM each primer, 2.5 U Taq DNA Polymerase (all reagents were supplied by Sigma-Aldrich, Saint Louis, MO, USA), 40–80 ng DNA, in a 50-μL final volume. PCR products were purified and sequenced at Macrogen Europe Laboratory (Madrid, Spain). The results were assembled, proofread, and edited using Sequencer 5.0 (Gene Code Corporation, Ann Arbor, MI, USA). The newly generated sequences were compared using BLASTn analyses (default settings) to those available in public nucleotide databases provided by the NCBI (Bethesda, MD, USA) and by the Westerdijk Fungal Biodiversity Institute (Utrecht, The Netherlands). Similarity values equal or higher than 98% were considered reliable. Fungi were preserved at the *Mycotheca Universitatis Taurinensis* (MUT) of the University of Turin. Newly generated sequences were deposited in GenBank (Accession Numbers from OP121168 to OP161816). Throughout the manuscript, fungal nomenclature follows the MycoBank database (Accessed on 23 August 2022: http://www.mycobank.org/).

### 2.5. Statistical Analysis

The biodiversity within sampling sites and matrices was evaluated by calculating the Shannon-Weaver index (H’), the Simpson index (1-Lambda), and the Pielou’s evenness (J’). For each matrix, significant differences among sub-sites and among sampling sites were evaluated by applying the permutational multivariate analyses of variance (PERMANOVA) and visualized by the canonical analysis of principal coordinates (CAP). The contribution of single species (in percentage) to the diversity observed within and between groups was assessed by similarity percentage (SIMPER) analysis. Statistical analyses were performed using PRIMER (Plymouth Routines in Multivariate Ecological Research, Albany Auckland, New Zealand) v 7.0 for multivariate ecological research [42].

## 3. Results

### 3.1. Charaterization of Sediments and MPs 

The granulometric analyses (Appendix A) showed that LH is dominated by fine grain/pelite sediments (ca 90%), while MPA are mainly formed by coarse/very coarse sand (82.06%) and gravel (16.52%). Finally, IP sediments mainly consisted of medium and fine grain sand (73.8%).

Appendix A report, respectively, the quantification of trace elements and organopollutants. LH showed higher levels of trace and organic elements. Regarding the trace elements, LH displayed a high level of Ni (42.77 mg/kg), Cu (48.86 mg/kg), and Zn (248.50 mg/kg); in comparison to IP and LH, MPA showed a higher level of As (30.55 mg kg). While organopollutants detected in MPA and IP were under the limit of quantification, LH was rich in PAHs and PCBs.

As for the MPs, in total, 292 items were retrieved: 88 (30%) from LH, 107 (37%) from IP, and 97 (33%) from MPA. Overall, the majority of the MPs were filaments (about 93 % of the characterized MPs, Appendix A). A small percentage were fragments (5%) or films (2%). IP showed a greater presence of filaments (33.3 ± 4.0 items/kg), followed by MPA (31 ± 11.0 items/Kg) and LH (26.3 ± 10.6 items/kg). The highest numbers of fragments were retrieved in LH (2.3 ± 3.2 items/kg), followed by IP (2.0 ± 1.7 items/kg) and MPA (0.7 ± 1.2 items/kg). As for films, IP showed 1.0 ± 1.7 items/kg, MPA 0.7 ± 1.2 items/kg, and IP 0.3 ± 0.6 items/kg. Styrofoam items were found only in IP (0.3 ± 0.6 items/kg). Regarding the colors, the main categories were blue (31%) < black (23%) < white transparent (18%) (Appendix A). Black items were dominant in MPA (11.0 ± 5.6 items/kg), while the blue ones prevailed in IP (11.3 ± 3.5 items/kg) and MPA (11.0 ± 5.3 items/kg). White transparent items were mainly retrieved in IP (11.3 ± 5.0 items/kg).

### 3.2. Fungal Diversity 

All samples were colonized by fungi. The SS colonization rates ranged between 280 ± 60 CFU g^−1^ dw (MPA) and 7580 ± 1260 CFUg^−1^ dw (LH). In WS, the highest fungal load was retrieved in LH samples (1512 ± 845 CFU 100 mL^−1^), followed by IP (1260 ± 349 CFU 100 mL^−1^) and MPA (187 ± 16 CFU 100 mL^−1^) (Appendix A). The biodiversity indexes are shown in Table 1. As for both Pielou’s evenness index (J’) and Simpson index (1-Lambda), the highest values were retrieved in MPs collected in IP and MPA. The highest Shannon–Wiener diversity index (H’) was observed in SS (the MPA SS had the highest H’ index: 3.89), followed by WS and MPs.

Overall, 1526 isolates representative of 315 taxa were retrieved (Appendix A): 779 from SS, 694 from WS, and 53 from MPs. Eighty-five percent of the taxa were identified at species level, 10% at genus, 1.5% at family, 1% at order, and 0.5% at class, while 1.5% remained identified at phylum level.

As illustrated in Figure 2, most of the isolated fungi (86%) were selectively associated to a specific matrix: 180, 85, and 7 taxa were exclusive to SS, WS, and MPs, respectively. Thirty-three taxa were shared between SS and WS, three between WS and MPs, and two between SS and MPs. Only five taxa were common to the three matrices. 

The phylum Ascomycota (91.7%) was, by far, the most represented, followed by Basidiomycota (6.3%), Mucoromycota (1.6%), and Mortierellomycota (0.3%). No Chytridiomycota or Cryptomycota were observed. The isolated taxa were ascribable to 15 classes, 35 orders, 70 families, and 120 genera. Among Ascomycota, the most represented classes were Sordariomycetes (38.4%; Hypocreales, Microascales, and Xylariales—64.9%, 9.9%, and 8.1%, respectively), Eurotiomycetes (33.9; mainly Eurotiales—93.9%) and Dothideomycetes (17.3%; mainly Pleosporales and Cladosporiales—54% and 36%, respectively), while Leotiomycetes (4.5%) and Saccharomycetes (3.1%) accounted for only a small part of the detected taxa. A low percentage (0.3%) of the Ascomycota remained unidentified. As for Basidiomycota, the most retrieved taxa belonged to Tremellomycetes (40%), Microbotryomycetes, Agaricomycetes, and Cystobasidiomycetes (15% each). All Mucoromycota belonged to the order Mucorales, while the single representative of Mortierellomycota was *Mortieriella alpina* (Mortieriellales). 

Most of the taxa (about 70%) were isolated on a specific medium, indicating the need for specific growth requirements (Appendix A).

### 3.3. SS Mycobiota 

Overall, 220 fungal taxa were isolated from SS, 180 of which were exclusive to that matrix. The fungal community was dominated by Ascomycota (97.3%): Sordariomycetes (39.1%), Eurotiomycetes (29.1%), Dothideomycetes (17.7%), Saccharomycetes (4.5%), Leotiomycetes (3.2%); a low percentage (0.5%) of Ascomycota remained unidentified. For Basidiomycota (2.7%), the retrieved taxa were affiliated to Agaricomycetes (1.4%), Cystobasidiomycetes, Exobasidiomycetes, and Ustilaginomycetes (0.5% each). 

As shown in Figure 3, *Trichoderma harzianum* resulted as the most abundant species (RA%, 24.01) in LH, followed by *Acrostalagmus luteoalbus* (RA% 10.40) and *Penicillium simplicissimum* (RA% 8.93). *Niesslia tenuis* (RA% 25.03) and *Penicillium chrysogenum* (RA% 20.05) dominated the IP sample site, followed by *T. harzianum* (RA% 3.28). On the contrary, the MPA mycobiota lacked dominant species, being characterized by many species with low relative abundance; e.g., *Cladosporium* sp.1 (RA% 5.82), *Purpureocillium lilacinum* and *Wardomicopsis humicola* (RA% 5.41 both), *Cladosporium perangustum* (RA% 4.96), and *Gliomastix murorum* (RA% 4.15). Overall, most of the taxa present in the three sampling sites showed a RA <3% (Appendix A).

### 3.4. WS Mycobiota

One hundred-twenty-six taxa (85 exclusive) were isolated from WS. Most of them were Ascomycota (87.3%), followed by Basidiomycota (7.9%). Mucoromycota (4%) and Mortierellomycota (0.8%) were poorly retrieved. Taxa mainly belonged to the classes Eurotiomycetes (35.7%), Sordariomycetes (28.6%), and Dothideomycetes (16.7%), followed by the Basidiomycota class Tremellomycetes (5.6%).

*Penicillium osmophilum* and *Rhodotorula diobovata* (RA% 45.18 and RA% 23.84, respectively) dominated in LH (Figure 4, Appendix A). As regards IP, the most abundant species were *R. diobovata* (RA% 21.71) and *Penicillium glandicola* (RA% 18.89), followed by *T. harzianum* (RA% 7.01) and *Penicillium hordei* (RA% 6.92). The species *Penicillium crustosum*, *Cladosporium halotolerans,* and *Penicillium manginii* (RA% 38.48, RA% 24.28, RA% 11.96, respectively) dominated the MPA sampling sites. 

### 3.5. MPs Mycobiota

Fifty-three fungi belonging to 17 taxa were isolated from the MPs (Table 2). Of these, seven (41.17%; Aspergillus domesticus, Cystobasidium slooffiae, Kondoa aeria, Penicillium bialowiezense, Sakaguchia dacryoidea, Sesquicillium microsporum, Vishniacozyma carnescens) were exclusive to MPs (Figure 2). At phylum level, the fungal community of the MPs consisted of Ascomycota (70.6%) and Basidiomycota (29.4%). Among the Ascomycota, the most represented were Dothideomycetes (29.4%), Eurotiomycetes (29.4%), and Sordariomycetes (11.8%), while those belonging to Basidiomycota were Cystobasidiomycetes (11.8%), Agaricomycetes (5.9%), Agaricostilbomycetes (5.9%), and Tremellomycetes (5.9%). As shown in Table 2, 12 taxa (70.5%) were isolated only with one method (sonication, 75%; direct plating, 25%). 

### 3.6. Fungal Community Structures

PERMANOVA analysis was used to test the differences in fungal community, within each sample sub-site and site for SS and WS, and for each site for MPs. For both SS and WS, the three sites showed a significantly different mycobiota (*p* < 0.05; Figure 5). Indeed, within each sampling site, the mycobiota of each sub-site was significantly different (*p* < 0.05), with the sole exception of LHA vs LHB (*p* > 0.05) for SS. 

Furthermore, the fungal communities associated with MPs collected in the three sites were significantly different, with the exception of IP vs MPA (*p* > 0.05; Figure 5).

As shown in Table 3, SIMPER analysis was applied to evaluate the dissimilarities among the different sites in the three matrices. As regards SS, the highest dissimilarity values were between LH vs MPA, followed by IP vs MPA and LH vs IP. In WS, the highest dissimilarity values were between IP vs MPA, followed by LH vs IP and LH vs MPA. As for MPs, IP vs MPA showed a dissimilarity of 98.61%, followed by LH vs MPA and LH vs IP. 

## 4. Discussion

Today, research on marine fungi is becoming more and more important. Indeed, beside their ecological relevance in the marine environments, these organisms could be potentially exploited as a source of new bioactive compounds or in bioremediation [43,44,45,46,47]. Therefore, studying the cultivable mycobiota of sediments (including watery sediments and associated microplastics) in both protected and contaminated sites is of fundamental importance. To the best of our knowledge, this is the first study to evaluate the fungal communities of different matrices, and along a transept with an anthropogenic impact gradient: from a polluted port, to a marine protected area.

Fungal isolation was performed using different techniques and media. The high number of taxa recorded was certainly due to the isolation procedures, since most of them were isolated in a single medium (or method), indicating specific growth requirements and/or morphophysiological peculiarities (i.e., lack of propagules). This indicates that the use of more than one medium allowed a better description of the fungal communities, as also described for deep-sea sediments [48] or other marine habitats [49,50]. The surprisingly high number of taxa observed (315) in this investigation was most likely also due to the number of replicates. Interestingly, 110 (34.9%) of them had never previously been detected in oceans (Appendix A).

### 4.1. Fungi from SS

Sediments represent an optimal habitat for marine fungi; the highest number of taxa isolated (200) was indeed retrieved from this matrix. The highest sediment colonization rates were in LH samples (7580 CFU ± 1260 g^−1^ dw). Cecchi et al. [51] studied culturable mycobiota of the sediment in six Mediterranean harbors (including the Livorno harbor) and observed a fungal load of 1370 CFU. Although LH sediments displayed the highest colonization rate, they showed the lowest diversity (Table 1). These results are in contrast with those of Khudyakova et al. [52], who found maximum values of fungal diversity in sediments with the highest anthropogenic impact. Our results could be explained by the high pollution detected in LH (trace elements: e.g., Ni, Cu or Zn; and organic PAHs: e.g., Benzo[a]pyrene or Benzo[b]fluoranthene). In this case, we can hypothesize that a fungal community is “shaped” to tolerate high levels of contaminants.

The SS mycobiota were dominated by Ascomycota, followed by Basidiomycota. This does not come as a surprise, since Ascomycota dominate coastal [53,54] and deep-sea sediments [48,55,56]. In this study, no Chytridiomycota were isolated, supporting the evidence that they usually cannot be detected by culture-dependent techniques. Indeed, using a culture-independent method, Rubin-Blum et al. [57] showed that Chytridiomycota was the most abundant phylum in Mediterranean nearshore sediments. This indicates that the use of both techniques for the assessment and understanding of the fungal community is necessary.

SS mycobiota were dominated by taxa belonging to the classes Sordariomycetes (39.1%), Eurotiomycetes (25.1%), and Dothideomycetes (17.7%), in line with previous reports [52,53,54]. The most abundant genera were *Penicillium* (13.2%), *Aspergillus* (7.3%), *Trichoderma* (7.3%), *Talaromyces* (5%), and *Cladosporium* (5%). Several investigations reported [44,45] these genera as the most abundant in marine sediments [40,46,51,54,58,59,60,61,62]. The same genera are also frequently found in fungal communities recovered from oil polluted sites such as LH [40,46,51,59,62], confirming the structure of the mycobiota of contaminated port sediments, including the presence of different *Aspergillus* spp. pathogens (e.g., *A sydowii* or *A. fumigatus*) [52,63]. 

Moreover, five Ascomycota isolates (three belonging to the genus *Massarina* and two to the order Pleosporales) remained sterile in axenic culture. Thus, morphological identification was not possible and molecular analyses did not lead to a clear conclusion. Therefore, these isolates deserve deeper investigations, and the possibility that they could be representatives of novel lineages cannot be excluded.

The SS fungal communities of the three sampling sites were significantly different. These results are not surprising, due to the marked physical and chemical differences of the sediments (Appendix A). SIMPER analyses (Table 3) highlighted that *P. simplicissimum* and *T. harzianum* contributed most to the differences observed. This is not unusual in marine environments, where the highly sporulating species of *Penicillium* and *Trichoderma* are frequent inhabitants [48,64,65]. These two species were the most abundant (RA%) in LH sampling sites, with *A. luteoalbus* and *S. chlorohalonata*. *P. simplicissimum* has previously been cultured from coastal and deep-sea sediments [40,48], while marine strains of *T. harzianum* have been engaged in bioremediation strategies of harbor sediments. *S. chlorohalonata* has been previously reported in harbor sediments [40], while *A. luteoalbus* has been found in the marine environments but never in port sediments [66,67]. As for IP, *N. tenuis* and *P. chrysogenum* dominated the mycobiota. *N. tenuis* was isolated for the first time in these habitats, probably due to its recent description [68]. Till now *N. tenuis* has been described as a common mycoparassite and highly sporulating species occurring on decaying fungal fruiting bodies of wood-inhabiting basidiomycetes; its ecological role in marine sediment is still unclear. In contrast, *P. chrysogenum* is a well-known inhabitant of marine sediments [12,40]. Regarding MPA, the RA% values ranged from 0.5 to 5.82, indicating that no species clearly dominated the fungal community. This result was confirmed by both the Shannon and Simpson indexes, which showed a higher evenness and species richness in MPA compared to LH and IP (Table 1). Among those with the highest values, *P. liliacinum, G. murorum* and species belonging to the genera *Cladosporium* have been retrieved in many different marine environments [67,69,70], while *W. humicola* was recently isolated for the first time on the brown alga *Padina pavonica* [41].

### 4.2. Fungi from WS

WS hosted a broad fungal community (126 isolated taxa), sharing only about 30% of taxa with the solid component of the sediments (Figure 2). The highest fungal load (1.512 CFU 100 mL^−1^) was retrieved in LH. As the distance from the coastline increased, a decrease of the fungal load was observed (1260 and 187 CFU 100 mL^−1^ in IP and MPA, respectively). This gradient could be explained by the proximity to the littoral zone (higher availability of nutrients). 

Similarly to SS, Ascomycota (87.3%) dominated this matrix, followed by Basidiomycota (7.9%). Mucoromycota (4%) and Mortierellomycota (0.8%), not observed in SS, were retrieved here, and have previously been reported in water environments [4,53]. Eurotiomycetes (35%), Sordariomycetes (28.6%), and Dothideomycetes (16.7%) were the most abundant classes, in line with previous studies that reported the prevalence of members of these classes in mangroves and coastal waters [71,72]. The most abundant genera were: *Penicillium* (31.75%), *Cladosporium* (11.90%), *Trichoderma* (7.14%), and *Fusarium* (5.56%). These genera are usually reported in marine environments [40,73,74], are highly sporulating, and species of these genera spread easily across the sea and their presence “hides” low sporulating fungi, as demonstrated by Shannon and Simpson indexes (Table 1). 

Tremellomycetes (5.6%) were also abundant. These basidiomycetous yeasts also recur in marine habitats and represent the majority of the yeast communities of the oceans [75,76]. The number of yeasts is higher in the shallow water sediments in comparison to the water column [69]. Among Tremellomycetes, three different red yeast taxa: *R. diobovata*, *R. mucillaginosa,* and *R. sphaerocarpa* were isolated. A high number of red yeasts is often correlated with water pollution, suggesting these organisms as indicators of water pollution [77,78]. Our results confirmed this assumption: the three species were mainly retrieved in LH. The RA% of *R. diobovata* strongly decreased from LH (23.84%) to MPA (0.53%) (Figure 4 and Appendix A). Moreover, a marine strain of *R. mucillaginosa* showed the capability to degrade dimethyl phthalate esters (DMPE), mainly used in plastic products. Paluselli et al. [79] reported a high concentration of DMPE in Mediterranean coastal areas, justifying the presence of *Rhodotorula* spp. in WS polluted sampling sites and suggesting their possible role in the degradation of the pollutant. However, LH is dominated not only by *R. diobovata* but also by *P. osmophilum* (RA = 45.18%). Despite the high RA of these taxa, to the best of our knowledge, this is the first time that *P. osmophilum* has been detected in a marine environment. In fact, *P. osmophilum* is normally retrieved in agricultural soils; however, the proximity to the mainland and the high anthropic impact can justify its high presence in LH. In both IP and MPA, different species of *Penicillium* (*P. glandicola*, *P hordei*, *P. canescens* for IP, and *P. crustosum*, *P manginii* for MPA), *Cladosporium* (*C. halotolerans* and *C. ramotenellum* for MPA), and *T. harzianum* were the most abundant. Most of these taxa are highly adapted to surviving and colonizing in marine water with peculiar chemical-physical conditions. However, the correlation of the high abundance of these genera to their extraordinary sporulation rate is evident; this may be a key of their colonization rate, or it could lead to overestimating the substrate colonization [80].

### 4.3. Fungi from MPs

Marine fungi remain under explored, which is also reflected by the limited number of studies focused on marine plastic fungal colonization [81,82]. Our results indicate that the cultivable mycobiota of the plastisphere within the sediments is dominated by Ascomycota (70.6%) and Basidiomycota (29.4%). However, despite the high number of fungi isolated, we must consider that the number of taxa (17) was much lower in comparison to the other two matrices within which the MPs were collected (220 and 126, for SS and WS, respectively). So far, only a few studies, using culture-independent methods, have investigated the fungal communities of the plastisphere in marine environments [83,84,85]. De Tender et al. [85] conducted a 44-week experiment during which polyethylene (PE) sheets were placed into the sediment at a harbor and an offshore location in the North Sea; they found that fungal communities were mainly represented by Ascomycota and Basidiomycota, while Mucoromycota were a minor fraction, in line with our results. In contrast, Kettner et al. [83,84] reported that the plastics mycobiota were dominated by Chytridiomycota and Cryptomycota, normally retrieved only by culture independent techniques.

Among the 17 taxa isolated in this study, five were yeast-like fungi: *Aureobasidium pullulans*, *C. slooffiae, S. dacryoidea*, *K. aeria*, *V. carnescens.* In marine environments, yeasts have a highly adaptable nature (e.g., hyperosmotic stress [86]), which can contribute to their ubiquitous distribution. In fact, they are known to be ecologically successful and to occupy a wide range of niches, due to their highly versatile physiological features [78,87]. Considering the theory of Baas Becking and Beijerinck, “everything is everywhere, but the environment selects” [88,89], we can assume that the plastisphere seems to be a preferential ecological niche for yeasts with respect to the surrounding sediments (both solid and watery parts) analyzed.

The MP fungal communities of the three sampling sites were significantly different (PERMANOVA; *p* ≤ 0.001), with the sole exception of IP vs MPA (*p* > 0.5). This tallies with other evidence: for example, Kettner et al. [84] showed that the location only, and not the type of polymer, significantly affected fungal communities from the Baltic Sea, River Warnow, and a wastewater treatment plant. 

Seven taxa out of 17 were exclusive of the MPs, indicating a matrix effect [83,84,85]. These differences suggest that plastic could affects the micro-eukaryotic community compositions of sediments sensu latu. Similarly, Oberbeckmann et al. [90] found significantly different fungal communities when comparing submerged plastic bottles, glass slides, and seawater in the North Sea. 

Among the exclusive taxa of MPs, we retrieved *A. domesticus*, a species of the *Aspergillus restrictii* section, that comprises xerotolerant species [91]. By studying the fungal diversity associated with plastics in the shallow waters of Antarctic peninsula and Atlantic Ocean, Lacerda et al. [92] found a prevalence of *Aspergillus vitricola*, *Aspergillus restricus*, and *Aspergillus wentii*, demonstrating that species belonging to *A. restrictii* section are important components of the plastisphere [91].

Finally, some of the taxa here identified may have the ability to degrade plastics. For example, *C. cladosporioides* and *C. pseudocladosporioides* were identified as potential degraders of PE and PU in terrestrial and aquatic environments [93,94]. Similarly, several strains of marine and terrestrial *Penicillium* and *Aspergillus* are active against different polymers [94,95,96,97,98,99]. Friedrich et al. [100], by screening 58 fungi, found that a strain of *Bjerkandera adusta* was the most promising for the degradation of Nylon-6. Therefore, fungi isolated from the plastisphere can be considered candidates for the biodegradation of MPs, as they are already adapted and capable of exploiting this new ecological niche. 

## 5. Conclusions

This work broadens the knowledge on both coastal marine sediments and microplastic-associated mycobiota, reporting for the first time the presence of several fungal species in the marine environment. Overall, a rich fungal biodiversity was found in the sediments, clearly split according to the different substrates, indicating how the matrix (SS, WS, MPs) deeply affects the fungal assemblage. On the other hand, the geographic location (and hence the anthropogenic impact) of the sampling sites is also a driving force in shaping the fungal underwater distribution. Fungal pathogens were found in sediments of the LH port; we cannot exclude that their presence derives from spreading and transport through MPs, as already stressed for pathogenic virus, bacteria, and protozoa. In conclusion, once more, this study underlined the importance of culturomics to the study of complex microbial ecosystems. Culturomics is a powerful tool that can complement the metabarcoding approach. The advantages of culturomics are (i) the isolation of fungi not detected by culture-independent methods; (ii) the isolation and description of putative new fungal species; (iii) the isolation in axenic cultures of fungal strains of biotechnological interest (e.g., production of bioactive compounds, degradation of xenobiotics, etc.). For this reason, all the fungal strains isolated and characterized in this study are preserved at the Mycotheca Universitatis Taurinensis (MUT) of the University of Turin. Indeed, in preserving microbial biodiversity, culture collections play a key role.

## Figures and Tables

**Figure 1 jof-08-00927-f001:**
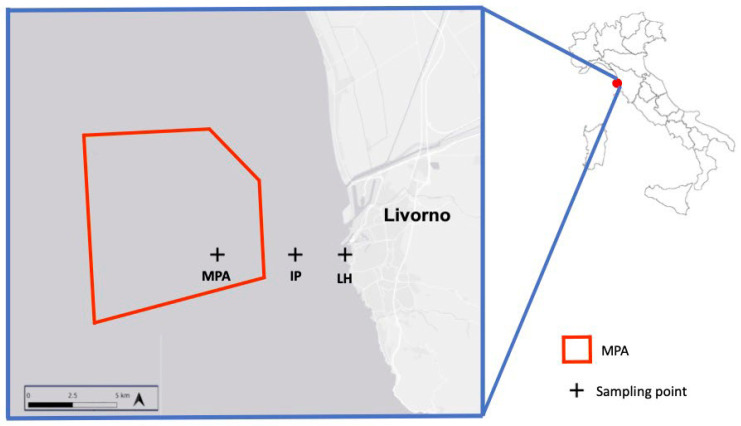
Map of the sampling sites: Livorno Harbor, LH; Intermediate Point, IP; Marine Protected Area “Secche della Meloria”, MPA.

**Figure 2 jof-08-00927-f002:**
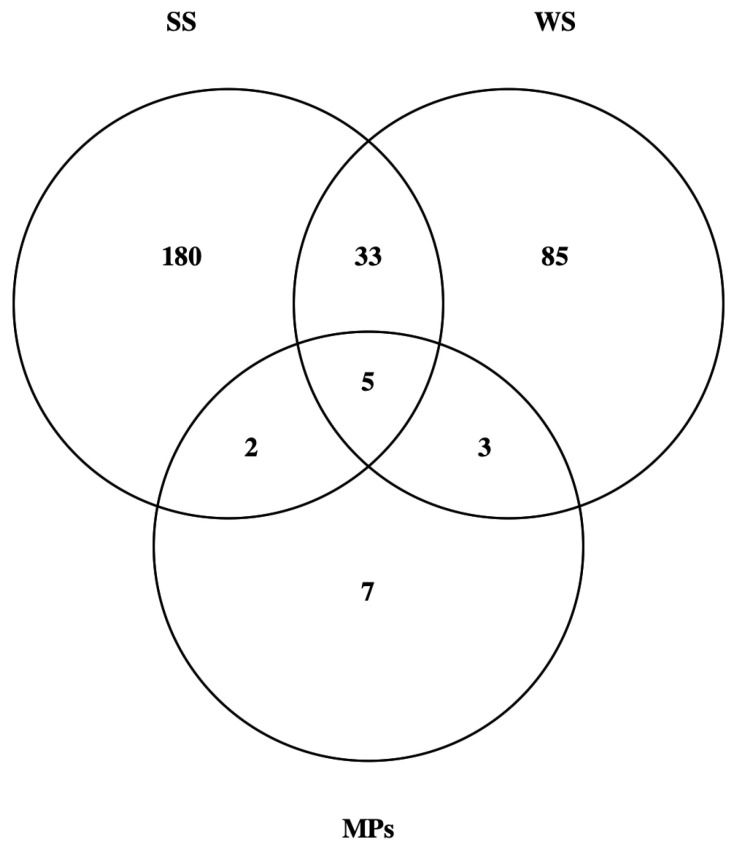
Venn diagram showing the number of exclusive and shared taxa among matrices: solid sediment (SS), watery sediment (WS), and microplastics (MPs).

**Figure 3 jof-08-00927-f003:**
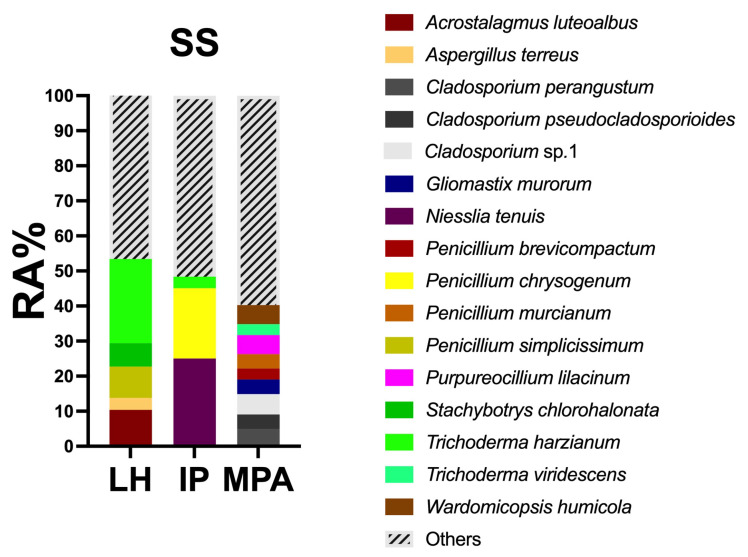
Relative abundance (RA%) of SS fungal taxa for Livorno harbor (LH), intermediate point (IP), and marine protected area (MPA). Fungal taxa with RA% < 3% are grouped in “others”.

**Figure 4 jof-08-00927-f004:**
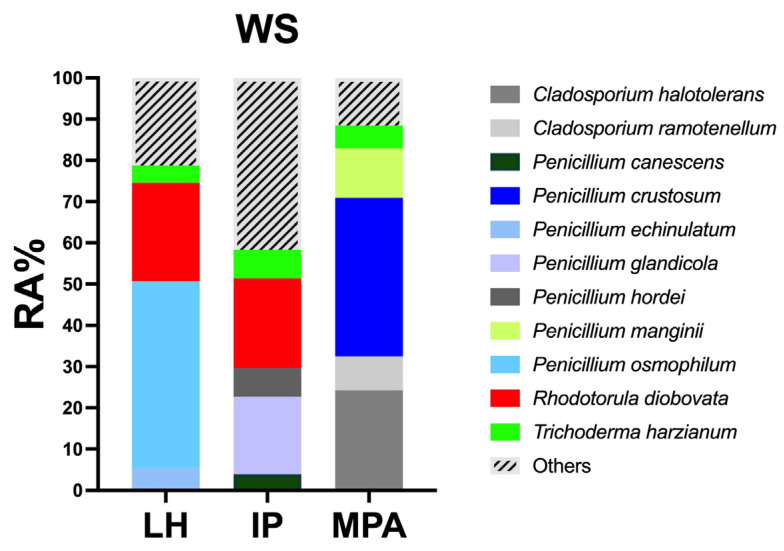
Relative abundance (RA%) of WS fungal taxa in Livorno harbor (LH), intermediate point (IP), and marine protected area (MPA). Fungal taxa with RA% < 3% are grouped in “others”.

**Figure 5 jof-08-00927-f005:**
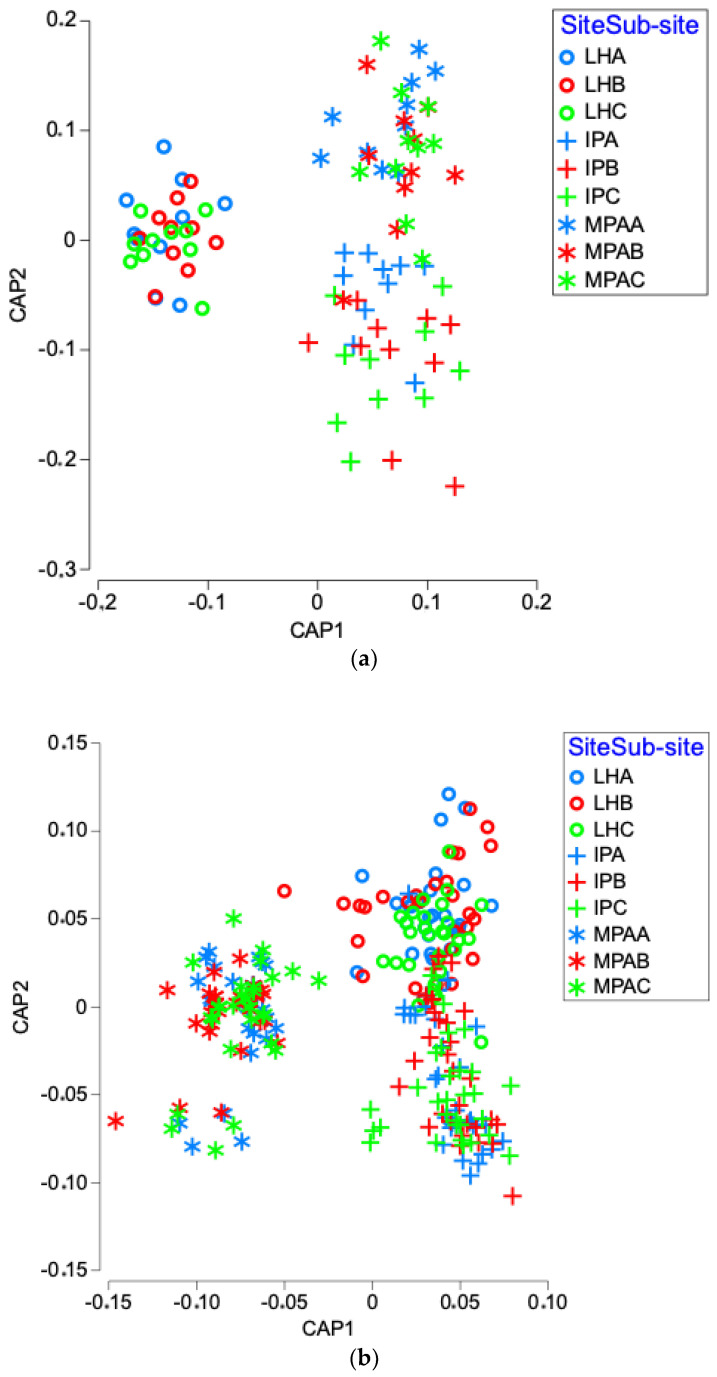
Canonical analysis of principal coordinates (CAP) illustrating the diversity of fungal communities (**a**) in solid sediment (SS); (**b**) watery sediment (WS), and (**c**) microplastics (MPs). In (**a**,**b**) symbols indicate Livorno harbor (LH), intermediate point (IP), and marine protected area (MPA), while colors indicate the different matrices within them.

**Table 1 jof-08-00927-t001:** Biodiversity indexes of sampling sites from different matrices.

Matrix	Site		Biodiversity Indexes *		
		S *	J’	H’	1-Lambda
SS	LH	88	0.74	3.32	0.91
	IP	123	0.69	3.35	0.89
	MPA	74	0.90	3.89	0.97
WS	LH	59	0.74	3.01	0.87
	IP	74	0.69	2.95	0.89
	MPA	58	0.49	1.98	0.77
MPs	LH	10	0.66	1.52	0.68
	IP	5	0.97	1.56	0.78
	MPA	9	0.95	2.09	0.86

* S = number of species; J’ = Pielou’s evenness; H’= Shannon index; 1-Lambda = Simpson index.

**Table 2 jof-08-00927-t002:** Fungal taxa isolated from MPs at Livorno harbor (LH), intermediate point (IP), and marine protected area (MPA), with different isolation methods: sonication or direct plating.

Taxa	MPs Isolation Methods				
	Sonication	Direct Plating	LH	IP	MPA
*Aspergillus domesticus*	x		x		
*Aspergillus jensenii*	x		x	x	
*Aureobasidium pullulans*	x		x		
*Bjerkandera adusta*		x			x
*Cladosporium cladosporioides*	x			x	
*Cladosporium halotolerans*	x	x	x	x	x
*Cladosporium pseudocladosporioides*	x		x		
*Cladosporium ramotenellum*	x	x			x
*Cystobasidium slooffiae*	x	x	x	x	x
*Kondoa aeria*	x				x
*Parengydontium album*	x		x		
*Penicillium bialowiezense*	x		x		
*Penicillium brevicompactum*	x	x	x		x
*Penicillium griseofulvum *		x			x
*Sakaguchia dacryoidea*	x	x		x	x
*Sesquicillium microsporum*	x		x		
*Vishniacozyma carnescens*		x			x

**Table 3 jof-08-00927-t003:** Dissimilarity of mycobiota among the three sites in different matrices. For each couple, the three taxa that mainly contributed to the dissimilarity are reported.

Matrix	Site	Dissimilarity %	Taxa
SS	LH vs IP	90.06	*Penicillium simplicissimum*
			*Stachybotrys chlorohalonata*
			*Acrostalagmus luteoalbus*
	LH vs MPA	95.23	*Penicillium simplicissimum*
			*Acrostalagmus luteoalbus*
			*Trichoderma harzianum*
	IP vs MPA	93.55	*Trichoderma harzianum*
			*Cladosporium* sp.1
			*Wardomycopsis humicola*
WS	LH vs IP	74.6	*Penicillium murcianum*
			*Penicillium griseofulvum*
			*Trichoderma harzianum*
	LH vs MPA	71.95	*Penicillium antarcticum*
			*Cladosporium halotolerans*
			*Rhodotorula diobovata*
	IP vs MPA	81.92	*Penicillium antarcticum*
			*Cladosporium halotolerans*
			*Penicillium griseofulvum*
MPs	LH vs IP	97.18	*Parengyodontium album*
			*Cladosporium halotolerans*
			*Cystobasidium sloffiae*
	LH vs MPA	98.17	*Parengyodontium album*
			*Cladosporium halotolerans*
			*Cystobasidium sloffiae*
	IP vs MPA	98.61	*Kondoa aeria*
			*Cladosporium halotolerans*
			*Cladosporium cladosporioides*

## Data Availability

Not applicable.

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
