# Peer review of "The Culturable Mycobiota of Sediments and Associated Microplastics: From a Harbor to a Marine Protected Area, a Comparative Study"

_jof, 2022, doi:10.3390/jof8090927_

Round 1

Reviewer 1 Report

Wonderful work. Congratulations. I have only a few minor issues that I would like to see addressed:

Line 19: “and threaten the organisms present in in the oceans” I’m not comfortable with this statement. I’m a chemist by original training and plastics are known in that field for being inert/non-reactive. They might be modified naturally or artificially to less inert molecules but such a hard statement of harmfulness has never been proved (to my knowledge). Please use a reference in the introduction to back this up or remove entirely.

Lines 42-43: “while the mycobiota associated with more accessible shallow-42 water sediments, such as those of coastal areas, are poorly investigated [2,11].” May I suggest that you add a recent paper that addressed this (and also coastal sand) by a broad team in Europe and another in Sydney, AU: http://dx.doi.org/10.1016/j.scitotenv.2021.146598

Lines 130-134: “To maximize the number of cultivable 130 fungi, isolation was carried out using three culture media: Malt Extract Agar (MEA: 20 g 131 L-1 malt extract, 20 g L-1 glucose, 2 g L-1 peptone, 18 g L-1 agar); Corn Meal Agar (CMA: 17 132 g L-1 CMA extract, 18 g L-1 agar); Synthetic Nutrient-poor Agar (SNA: 1 g L-1 KH2PO4, 1 g 133 L-1 KNO3, 0.5 g L-1 MgSO4. 7H2O, 0.5 g L-1 KCl, 0.2 g L-1 glucose, 0.2 g L-1 saccarose, 18 g 134 L-1 agar).” I don’t understand this. I’ve been working in Mycology for decades and we use a rich medium such as Sabouraud to have a ‘maximum recovery’, not poor media, which are used for identification of structures produced often only in such medium. You start the sentence stating that you want to maximise the number of cultivable fungi and then describe 3 poor media instead of a rich one. Was this done to avoid first isolating in rich medium and then reisolating individual colonies in poor medium for the identification structures? Please clarify clearly in the text.

Author Response

Response to Reviewer 1 Comments

Dear Editor,

I wish to thank you for giving us the opportunity to submit a revised version of the manuscript entitled "The culturable mycobiota of sediments and associated microplastics: from a harbour to a marine protected area, a comparative study" to Journal of Fungi. We appreciate the time and effort that you and the reviewers dedicated to provide valuable feedback. We are grateful to the reviewers for their insightful comments. Changes highlighted in the manuscript reflect the reviewers’ suggestions.

Point-by-point responses to the reviewers’ comments and concerns are hereunder reported.

Point 1: Line 19: “and threaten the organisms present in in the oceans” I’m not comfortable with this statement. I’m a chemist by original training and plastics are known in that field for being inert/non-reactive. They might be modified naturally or artificially to less inert molecules but such a hard statement of harmfulness has never been proved (to my knowledge). Please use a reference in the introduction to back this up or remove entirely.

Response 1: I inserted a reference in the introduction (L 62), as microplastics (plastics in general), despite their inert nature, could carry pollutants (including persistent organic pollutant), contaminate marine organisms (if ingested), thus affecting the trophic chain.

Point 2: Lines 42-43: “while the mycobiota associated with more accessible shallow water sediments, such as those of coastal areas, are poorly investigated [2,11].” May I suggest that you add a recent paper that addressed this (and also coastal sand) by a broad team in Europe and another in Sydney, AU: http://dx.doi.org/10.1016/j.scitotenv.2021.146598

Response 2: done

Point 3: Lines 130-134: “To maximize the number of cultivable 130 fungi, isolation was carried out using three culture media: Malt Extract Agar (MEA: 20 g 131 L-1 malt extract, 20 g L-1 glucose, 2 g L-1 peptone, 18 g L-1 agar); Corn Meal Agar (CMA: 17 132 g L-1 CMA extract, 18 g L-1 agar); Synthetic Nutrient-poor Agar (SNA: 1 g L-1 KH2PO4, 1 g 133 L-1 KNO3, 0.5 g L-1 MgSO4. 7H2O, 0.5 g L-1 KCl, 0.2 g L-1 glucose, 0.2 g L-1 saccarose, 18 g 134 L-1 agar).” I don’t understand this. I’ve been working in Mycology for decades and we use a rich medium such as Sabouraud to have a ‘maximum recovery’, not poor media, which are used for identification of structures produced often only in such medium. You start the sentence stating that you want to maximise the number of cultivable fungi and then describe 3 poor media instead of a rich one. Was this done to avoid first isolating in rich medium and then reisolating individual colonies in poor medium for the identification structures? Please clarify clearly in the text.

Response 3: Based on our experience (https://doi.org/10.1016/j.scitotenv.2016.10.064; 10.3390/md17040220; 10.1016/j.nbt.2013.01.010; https://doi.org/10.1111/1462-2920.15560) the use of media that differ in the content of C and N can maximize the isolation of fungi with diverse nutritional needs. MEA and CMA are rich media with different C and N sources and ratio and SNA is a poor medium. The use of different culture media here has not been adopted for morphological identification, but to prevent highly sporulating species from covering species that grow slower or require less rich media for growing (marine sediments and water are considered oligotrothic environments). Indeed, in the discussion we point out that none of the media used maximises the number of cultivable fungi; nontherless the synergic use of three media allowed the isolation of a high number of fungal taxa.

Reviewer 2 Report

The authors investigated microplastic classes, trace elements and chemicals in sediments of 3 marine sites. The sites were differently impacted by human activities, including a harbor and a protected area as well as a site in-between. The authors did an impressing approach to isolate fungi from the different plastic and sediment particle classes. They isolated more than 1500 strains comprising 315 taxa. Substrate specificity and sampling site shaped the fungal community as documented by tables and +/- statistics. Isolating fungi is laborious, needs expertise, but is rewarding in terms of future experiments concerning e.g. plastic degrading abilities and last but not least taxonomics. This tremendous approach to investigate fungi on marine microplastic qualifies this paper to be published in JoF. Discussion is thorough in most aspects and interesting to read with valuable information about yeast-like fungi and their adaptations to life on microplastics. The conclusions are sound.

Unfortunately, the manuscript is carelessly written in some sections. Tables are not in order, two tables are missing, x-axis label is wrong in Figure S1. I indicated many minor wording mistakes directly on the PDF version of the manuscript. In particular the first results section is difficult to read due to language issues. Please ask an English language expert to thoroughly revise your manuscript. Please check you reference list.

Specific comments (also on PDF file)

L 135/136 Please explain the use of Gentamycin and Tazobactam (e. g. instead of Chloramphenicol), is this related to isolation of marine Fungi in particular?

L 165 markers table is missing

L 194 wrong table here, information is in Table S1 instead of S2

L 256 replace IT by IP in Figure S1 x-axis label

L 347-363 The sampling success is most probably achieved by methods and media applied. However, the Venn diagram showed also the high substrate specificity of the fungi isolated and thus the little overlap between samples. Did you test your sampling coverage? This would certainly be interesting to add. Table S 8 is missing!

L 378 Could the use of other culture media specifically targeting Chytridiomycota (some are able to grow axenic, or are saprotroph) solve this??

Author Response

Response to Reviewer 2 Comments

Dear Editor

I wish to thank you for giving us the opportunity to submit a revised version of the manuscript entitled "The culturable mycobiota of sediments and associated microplastics: from a harbour to a marine protected area, a comparative study" to Journal of Fungi. We appreciate the time and effort that you and the reviewers dedicated to provide valuable feedback. We are grateful to the reviewers for their insightful comments. Changes highlighted in the manuscript reflect the reviewers’ suggestions.

Point-by-point responses to the reviewers’ comments and concerns are hereunder reported.

I apologize for the errors related to English and for the missing/not in the correct order tables. Before the submission, I found that the file I was working on was corrupt. I tried to fix the problem but I probably uploaded the file still damaged, which did not save the changes made to the revisions before submission.

Point 1: L 135/136 Please explain the use of Gentamycin and Tazobactam (e. g. instead of Chloramphenicol), is this related to isolation of marine Fungi in particular?

Response 1: The use of Gentamycin and Tazobactam is not only related to the isolation of marine fungi, but to the isolation of fungi in general in axenic culture. In our experience, we optimized the use of a mix of these two antibiotics to prevent bacterial growth. In fact, the use of other antibiotics (e.g. Chloramphenicol or Streptomycin) was not sufficient to avoid the growth of some bacteria commonly present in environmental samples.

Point 2: L 165 markers table is missing

Response 2: I deleted this table and inserted a quote, reporting all the molecular markers used in this study. The supplementary tables were modified, using the correct order.

Point 3: L 194 wrong table here, information is in Table S1 instead of S2

Response 3: done.

Point 4: L 256 replace IT by IP in Figure S1 x-axis label

Response 4: done

Point 5: L 347-363 The sampling success is most probably achieved by methods and media applied. However, the Venn diagram showed also the high substrate specificity of the fungi isolated and thus the little overlap between samples. Did you test your sampling coverage? This would certainly be interesting to add. Table S 8 is missing!

Response 5: Indeed, while in a metabarcoding approach it is relatively easy to calculate the sampling coverage through a rarefaction curve, the same is not true for a culturomic approach. We are not able to calculate a sampling coverage for the different matrices even referring to the subsample of cultivable fungi. What we have been able to carry out for the sediment analysis is a preliminary analysis aimed at calculating the most suitable dilution for each sample and the use of different media (different C and N source and different C/N ratio) to favor the isolation of fungi with different nutritional needs and growth rate. Regarding the analysis of water, on the basis of previous experiments (BOVIO, Elena, et al. The culturable mycobiota of a Mediterranean marine site after an oil spill: Isolation, identification and potential application in bioremediation. Science of the Total Environment, 2017, 576: 310-318.) we have already modified the protocol normally used by increasing the number of replicates and decreasing the quantity of filtered water (ml) for each replicate. Finally, for microplastics we used two sampling methods (water plating deriving from sonication vs direct plating of the sonicated plastic pieces) to favor the isolation of different fungi (sporulants, sterile mycelia, adhered to the substrate through the production of biofilm). However, it should be emphasized that the number of isolated taxa in this work is very high, and in general greater than that found in similar works.

Point 6: L 378 Could the use of other culture media specifically targeting Chytridiomycota (some are able to grow axenic, or are saprotroph) solve this??

Response 6: Thank you very much for the interesting comment. In future investigations, we will try to isolate Chytridiomycota using other isolation methods. As reported in literature (https://umaine.edu/chytrids/isolation-methods-for-chytrids/) the most common method to isolate Chytridiomycota from soil and water samples considers the use of bait (normally chitin bait). Following, once captured, the use of specific media such as PmTG, mPmTG, CMA or 1/4 strength YPSs is necessary for the preservation in axenic culture. Baits are indeed necessary for the isolation, since by using one of the media mentioned above- (CMA) we did not detect Chytridiomycota. In addition, we were not able to replicate anaerobic conditions in our lab, which is required for many Chytrids.Finally, many Chithridiomycota are mutualist or pathosistic symbiotes of other organism, therefore difficult to isolate under in conditions.

POINT 7 L387These genera are the most abundant in marine sediments or are they just easy to isolate?

Response 6: This point was discussed in L434-438. Indeed, these highly sporulating species dominate SS and WS.